# The Expressivity of Fixed-Precision Transformers without Positional Encoding

## Abstract

The primary objective of this study is to examine how practical constraints impact the expressivity of Transformers and to investigate their expressivity in real-world implementations.

To achieve this, we analyze the expressivity of Transformer decoders operating under fixed-precision float arithmetic, an assumption regarding query-key parameters, and the presence or absence of positional encoding. Our findings reveal that, under fixed-precision and these constraints, Transformers are limited to recognizing finite or co-finite languages, a proper subclass of regular languages. While incorporating positional encoding or relaxing certain assumptions marginally enhances expressivity, the fundamental limitations imposed by fixed precision remain significant.

These results underscore the gap between theoretical models and real-world implementations, suggesting that practical Transformers may be fundamentally constrained to recognizing only finite and co-finite languages, effectively functioning as little more than efficient lookup tables.

## 1. Introduction

The expressivity of Transformer models (Vaswani et al., 2017) has been further elucidated through recent theoretical analyses by comparing to the range of recognizable formal languages and solvable complexity classes. A series of studies has established upper and lower bounds on their expressivity under following settings.

Pérez et al. (2021) is the first study to explore the expressivity of Transformers, proving their Turing-completeness using rational numbers, assuming infinite precision float. Subsequent studies adopting finite precision have provided more practical insights. For instance, Merrill & Sabharwal

Table 1. The upper and lower bound of the expressivity of fixed-precision Transformers. Chiang et al. (2023)[†] identified the upper bound of normal Transformer encoder model (i.e. fixed-precision, sinusoidal positinoal encoding). In this study we showed that **bold** parts. "?" means the bound is not known.

| Asm. | Assumption. 5.1 | | – | |
| --- | --- | --- | --- | --- |
| PE | NoPE | APE | NoPE | APE |
| Uppoer bound | **FinCofin** (§ 5.1) | ? | ? | FOC [+; MOD][†] |
| Lower bound | **FinCofin** (§ 5.2) | **FinCofin** $m$-**cyclic** (§ 6.1) | **FinCofin letter-set** (§ 6.2) | ? |

(2023; 2024a) investigated logarithmic precision, which is finite but scales with input length $n$, and revealed that such Transformers are limited to much smaller circuit complexity classes, such as $TC^0$ or logical class $FO(M)$, compared to Turing machines. Similarly, Chiang et al. (2023) examined fixed-precision Transformers and demonstrated that their tighter upper bounds are linked to logic $FOC[+; MOD]$, which is an extension of first-order logic.

Despite these theoretical advances, many studies rely on idealized conditions. This paper bridges the gap between these settings and real-world implementations, which impose significant constraints on processing and retaining information.

We investigate how the expressivity of Transformer decoders is shaped by the following practical constraints: *fixed-precision floating-point numbers*, *positional encoding variations* (APE, NoPE), and *assumptions on parameter configurations* (asm. 5.1). Our results indicate that expressivity depends on these constraints as follows (Table 1).

- Fixed-precision (e.g., fp32, bf16) limits recognition to finite and co-finite languages. $\{a, b, ba, aab\}$ (§ 5)

- Absolute positional encoding extends recognition beyond finite and co-finite languages to cyclic languages. $\{ab, abab, ababab, \dots\}$ (§ 6.1)

- Non-finite values ($\pm\inf$) expand expressivity to letter-set languages, capturing specific letter inclusion. $\{abbb, ccac, bbac, \dots\}$ (§ 6.2)

Our findings extend prior results by highlighting the theoretical and practical implications of fixed-precision.

[1]Anonymous Institution, Anonymous City, Anonymous Region, Anonymous Country. Correspondence to: Anonymous Author <anon.email@domain.com>.

## 2. Related Work

### 2.1. Transformer Models and Expressivity

The computational capabilities of neural networks, covering RNNs, CNNs, and Transformers, have been extensively studied. The comprehensive surveys by Ackerman & Cybenko (2020); Merrill (2021; 2023) provide an in-depth overview of the expressivity of neural networks as a whole.

Learnability is inherently bounded by expressivity, as the language that a model can recognize defines the boundaries of what it can effectively learn. Therefore, expressivity is not only a theoretical concern, but it is also of practical importance in guiding model design.

A survey paper (Strobl et al., 2024) and lecture notes (Chiang et al., 2024) provide a comprehensive overview of recent advances in the study of Transformer expressivity, highlighting that expressivity is often analyzed in relation to three key areas: formal languages, circuit complexity, and logic.

**Formal languages**  Hahn (2020); Bhattamishra et al. (2020a;b); Yao et al. (2021); Chiang & Cholak (2022) primarily investigated the relationship between variants of hard-Transformers and formal languages such as PARITY and Dyck languages, which are commonly used benchmarks for expressivity. Feng et al. (2023); Merrill & Sabharwal (2024b) focused on the decoding time, inspired by chain-of-thought reasoning (Wei et al., 2022), demonstrating that expressivity expands significantly with multiple decoding steps. Of particular interest, Nowak et al. (2024) examined how Transformers assign probabilities to strings in language modeling, identifying connections to probabilistic deterministic finite automata and probabilistic Turing machines.

**Circuit complexity**  Another perspective comes from circuit complexity theory, which classifies computational problems based on their implementability within Boolean circuits of bounded depth and size. Hao et al. (2022) analyzed hard-Transformer variants, linking them to the tiny circuit class $AC^0$. Merrill et al. (2022); Merrill & Sabharwal (2023) extended this to more practical settings, showing that the saturated attention and logarithmic precision Transformers remain within $TC^0$. Merrill & Sabharwal (2023) further suggested a fundamental parallelism trade-off, arguing that highly parallel architectures like Transformers may inherently face computational limits.

**Logic**  Chiang et al. (2023); Merrill & Sabharwal (2024a); Yang et al. (2024); Yang & Chiang (2024) have explored connections between Transformer models and first-order logic. These studies encode strings into Boolean variables and represent languages using logical frameworks such as first-order logic with counting quantifiers ($FOC[+; MOD]$).

While significant progress has been made, many studies rely on unrealistic assumptions such as infinite precision or hard-attention, leaving questions about their practical relevance.

### 2.2. Neural Networks and Function Approximation

A fundamental result in neural network theory is the universal approximation theorem, which states that any continuous function can be approximated arbitrarily well. While not the focus of our study, it provides essential context for understanding the broader capabilities of neural networks.

**Feedforward networks**  Feedforward neural networks (FFNs) play a central role in this context. Cybenko (1989); Hornik et al. (1989) proved that FFNs with a single hidden layer and arbitrary nonlinear activations can universally approximate any Borel measurable or continuous function, given sufficient hidden units. Park et al. (2020) further identified the minimum width required for universal approximation, given the input and output dimensions.

**Transformers**  Recent work has extended universal approximation results to Transformers, with Yun et al. (2020) establishing their ability to approximate continuous sequence-to-sequence functions on compact domains and highlighting the crucial role of positional encoding in encoding order and circumventing permutation equivariance constraints. Kajitsuka & Sato (2024) later showed that even single-layer Transformers with low-rank weights can achieve such approximation power. Furthermore, Wei et al. (2022) introduced the statistically meaningful approximation framework, addressing limitations in classical approximation theory by incorporating learnability constraints.

## 3. Preliminaries

In this section, we present the foundational concepts that support our theoretical results. For strings $w, w' \in \Sigma^*$ over the alphabet $\Sigma$, $|w|$ denotes the length of the string, and $ww'$ denotes the concatenation. Furthermore, $w_t$ denotes the $t$-th character, and $w_{i:j}$ $(i, j \in \mathbb{N})$ denotes the subsequence of $w$ from the $i$-th to the $j$-th character.

### 3.1. Finite and Co-finite Languages, Cyclic Language, Letter-set Language

This subsection introduces *finite languages* and their dual, *co-finite languages*, along with *letter-set languages* and *cyclic languages*. These languages will play a central role in analyzing the expressivity of Transformers (§ 5, § 6).

**Definition 3.1** (Finite Language). Let $\Sigma$ be a finite alphabet. A language $L \subseteq \Sigma^*$ is called a *finite language* if and only if there exists $k \in \mathbb{N}$ such that for all stings $w \in L, |w| \leq k$.

**Definition 3.2** (Co-finite Language). Let $\Sigma$ be a finite alphabet. A language $L \subseteq \Sigma^*$ is called a *co-finite language* if and only if its complement $\Sigma^* \setminus L$ is a finite language.

**Definition 3.3** ($m$-cyclic Language). Let $\Sigma$ be a finite alphabet. A language $L \subseteq \Sigma^*$ is called a *$m$-cyclic language* if and only if for some $m \in \mathbb{N}$, for all $w, w' \in L$ and for all $0 \le i \le \max(|w|, |w'|)$, $w_i \equiv w'_i \mod m$ holds.

**Definition 3.4** (Letter-set Language). Let $\Sigma$ be a finite alphabet. A language $L \subseteq \Sigma^*$ is called a *letter-set language* if and only if for some set of letters $A \subseteq \Sigma$, for all $w \in L$ includes all of the letters in $A$.

**Example 3.5.** *The following languages $L, L'$ over $\Sigma = \{a, b\}$ are co-finite languages:*

$$L = \Sigma^* \setminus \{a, b, ab, aab\}$$
$$L' = \{w \in \Sigma^* \mid |w| \ge 3\}$$

*Similarly, the following language $L_{\{a,b\}}$ over $\Sigma = \{a, b, c\}$ is letter-set language and $L_3$ is 3-cyclic language:*

$$L_3 = (abc)^*$$
$$L_{\{a,b\}} = \{w \in \Sigma^* \mid w \text{ has both } a \text{ and } b\}$$

### 3.2. $p$-precision Float-Point Numbers

Now we define the rigorous mathematical framework for representing and manipulating numerical values under finite precision constraints, following (Merrill & Sabharwal, 2023).

**Definition 3.6** ($p$-precision Floating-Point Numbers (Merrill & Sabharwal, 2023)). The set of $p$-precision floating-point numbers $\mathbb{D}_p$ is defined as the collection of $p$-bit numbers, $\mathbb{D}_p = \{0, 1\}^p$, including special values such as $+\inf$, $-\inf$, and $\text{nan}$. The set $\mathbb{D}_p$ can be naturally extended to vectors $\mathbb{D}_p{}^*$.

When $p$ happens to be a finite number, we can also define the operations over $p$-precision float since the cardinality of the mappings between $\mathbb{D}_p$ vectors of $m$-dimension become at most finite ($= 2^{pm \cdot 2^{pm}}$).

**Definition 3.7** (Merrill & Sabharwal (2023)). A function $f : \mathbb{D}_p{}^m \to \mathbb{D}_p{}^n$ is a $p$-precision floating-point function if $f$ can be computed by a $p$-space-bounded Turing machine.

The order and basic operations, including addition, subtraction, multiplication, and division, as well as operations involving special values ($+\inf$, $-\inf$, $\text{nan}$), follow the IEEE 754 standard (iee, 2019).

The precision $p$ can be defined as a function of the input sequence length $n$, determining the scale of precision as follows:

- Constant Precision: When $p(n)$ is a constant function ($p(n) \in \mathcal{O}(1)$), the precision is fixed for any length of input.

- Logarithmic Precision: When $p(n)$ is a logarithmic function ($p(n) \in \mathcal{O}(\log n)$), the precision scales logarithmically with the input length.

In this work, our concern is constant precision. In the case of constant precision, $p$ can be treated as a constant $p \in \mathbb{N}$.

## 4. Transformer Decoder

This section introduces the mathematical and theoretic foundations of the Transformer Decoder Model, emphasizing its functional behavior (Def. 4.1), autoregressive capabilities (Def. 4.2), and alignment with formal languages (Def. 4.3).

### 4.1. Transformer Decoder

We focus on the decoder-based GPT (generative pretrained transformer) architecture (Radford et al., 2018). Unlike the original implementation, positional encoding (PE; details in § 4.1) is excluded in § 5 to facilitate theoretical analysis, while it is included in § 6 to reflect practical settings and evaluate its impact.

In this work, all computations within the Transformer are conducted over the $p$-precision float numbers $\mathbb{D}_p$ (see § 3.2 and Merrill & Sabharwal (2023)). This constraint reflects a practical adaptation to real-world computational limits.

**Vocabulary space**  The vocabulary space of Transformers $\Sigma \cup \mathbb{V}$ comprises the alphabet $\Sigma$ and a set of special tokens $\mathbb{V}$ (e.g., $\langle\text{bos}\rangle$, $\langle\text{eos}\rangle$, $\langle\text{sep}\rangle$). Special tokens are excluded from formal language, and there is no intersection to alphabet in this study. Basic string operations, such as concatenation, closure, and length, are defined over vocabulary spaces in the standard manner.

**Transformer as a function**  Then we formalize the Transformer Decoder model as a function.

**Definition 4.1.** A Transformer Decoder over $p(n)$-precision with parameters $\theta \in \text{Params}$ is a function:

$$\text{TDec}_p(\cdot; \theta) : (\Sigma \cup \mathbb{V})^* \to \Sigma \cup \mathbb{V}$$

where $\Sigma \cup \mathbb{V}$ is the vocabulary space. $\text{Params}$ represents the class of trainable parameters set, all components of the model. $p(n)$ determines the internal precision depend on the input sequence length $n$.

Given an input sequence $w_{1:n} \in (\Sigma \cup \mathbb{V})^*$, the Transformer Decoder outputs a single next token $w_{n+1} = \text{TDec}_{p(n)}(w_{1:n}; \theta) \in \Sigma \cup \mathbb{V}$, conditioned on the prefix $w_{1:n}$ and a set of parameter $\theta$. Based on the formal definition of TDec, the computational flow from input to output generally follows the GPT model (Radford et al., 2018; Brown et al., 2020).

**Positional encoding** Since encoder-only Transformer cannot recognize the position of character, they need additional positional information. We denote absolute positional encoding (APE) like Vaswani et al. (2017)'s sinusoidal one in this work. On the other hand, there are relative ones like T5 relative PE (Raffel et al., 2020) or ALiBi (Press et al., 2022). Alternatively, Kazemnejad et al. (2024) showed *No Positional Encoding (NoPE)* has good ability to generalize. In this work, we employ only APE and NoPE.

### 4.2. Autoregressive Token Generation

The Transformer Decoder model generates sentences autoregressively, predicting each token based on the input sequence and previously generated tokens until output a kind of end-of-sentence tokens. This process is formalized as follows:

**Definition 4.2.** The $t$-times autoregressive composition (generation) of the Transformer Decoder function $\mathrm{TDec}_p(\cdot; \theta)$ is denoted as $\mathrm{TDec}_p{}^t(\cdot; \theta) : (\Sigma \cup \mathbb{V})^* \to \Sigma \cup \mathbb{V}$ and is recursively defined as:

$$\mathrm{TDec}_p{}^t(\sigma; \theta) = \begin{cases} \mathrm{TDec}_p(\sigma \cdot \mathrm{TDec}_p{}^{t-1}(\sigma; \theta); \theta) \\ \qquad\qquad\qquad\qquad \text{(if } t > 1) \\ \mathrm{TDec}_p(\sigma; \theta) \\ \qquad\qquad\qquad\qquad \text{(if } t = 1) \end{cases}$$

where $\cdot$ denotes token concatenation over $\Sigma \cup \mathbb{V}$.

This definition highlights the iterative nature of autoregressive generation, Furthermore, by restricting the codomain to the last token, this formulation aligns with the objectives of this study, emphasizing the relationship between autoregressive behavior and formal language recognition. From now on, when the context is clear, we simply write TDec.

### 4.3. The Language Recognized by Transformer Decoder

We now define the language recognized by a $p$-precision Transformer Decoder with a certain parameter $\theta$ and $t$-times decode steps, based on the definition 4.1 and 4.2.

**Definition 4.3.** The language recognized by such a $t$-times autoregressive transformer decoder with a certain parameters $\theta$ over $p$-precision, $\mathrm{TDec}_p(\cdot; \theta)$, $L(\mathrm{TDec}_p(\cdot; \theta))$ is defined as:

$$L(\mathrm{TDec}_p{}^t(\cdot; \theta), F) \\ = \{w \in \Sigma^* \mid \exists r \leq t(|w|). \mathrm{TDec}_p{}^r(w \cdot \langle \mathrm{sep} \rangle; \theta) \in F\} \quad (1)$$

where $F \subseteq \mathbb{V}$ is the nonempty set of accept token. Typically, $F$ may include tokens such as $\langle \mathrm{eos} \rangle$ or other special markers representing accept tokens.

Definition 4.3 states that an input string $w$ is accepted if the output sequence $\mathrm{TDec}(w \cdot \langle \mathrm{sep} \rangle) \in (\Sigma \cup \mathbb{V})^*$ contains at least one accept token from $F$, within $t(|w|)$ times or less autoregression.

It is important to note that the special token $\langle \mathrm{sep} \rangle$ is explicitly appended to the input sequence to distinguish the decoding sequence. Additionally, the length of the output sequence increase by a time function $t : \mathbb{N} \to \mathbb{N}$, which maps the input sequence $w$ to a maximum allowable number of decoding steps. For example: If $t(n) = n^2$, polynomially many decoding steps are permitted. If $t(n) = c$, decoding is restricted to a constant steps, regardless of the input length.

**Example 4.4.** *Let the time function be a constant function $t(n) = 4$, and the set of accept tokens be $F = \{\langle \mathrm{eos} \rangle\}$. Given that the output sequences of* TDec *for the input sequences "aabb" and "aa" are as follows:*

$$\mathrm{TDec}(aabb\langle \mathrm{sep} \rangle) = aba\langle \mathrm{eos} \rangle \dots$$
$$\mathrm{TDec}(aa\langle \mathrm{sep} \rangle) = aaaa \dots$$

*In this case, the Transformer accepts only "aabb".*

### 4.4. Confirmation of Constraints

All other hyperparameters, such as the number of layers $L \geq 2$, the model dimension $d$, and attention heads, are fixed as $O(1)$, regardless of the input sequence length $n$. In summary, this study incorporates certain modifications:

- *Exclusion of positional encoding (NoPE; only for § 5)*
- Two-layer Transformer Block, Single-head Attention without pre-norm configuration (§ 4.1)
- Causal masking for attention computation, and softmax function within the Attention mechanism (§ 4.1)
- Greedy Search decoding (Definition 4.3)

This formalization bridges the autoregressive generation mechanism with the theoretical analysis of language recognition. In subsequent sections, we explore the expressivity of Transformer Decoder models within this framework.

## 5. Main Result 1: Finiteness of Fixed-Precision Transformer without PE

In this section, we present our first main result concerning the expressivity of Transformers under fixed-precision arithmetic and softmax-based attention mechanisms. This result establishes a direct correspondence between the class of languages recognized by Transformers and finite or co-finite languages (Theorem 5.2) under a natural assumption (Assumption 5.1).

**Infinity-Free Parameter Assumption** We begin by introducing a natural assumption regarding the parameters of the attention layers in Transformers.

**Assumption 5.1** (Infinity-Freeness)**.** For each attention layer, the matrix product of query and key vectors is always greater than minus infinity ($-\inf \in \mathbb{D}_p$):

$$\forall y, y' \in \mathbb{D}_p{}^d. \quad Q(y)K(y')^\top \neq \pm\inf \tag{2}$$

where $d \in \mathbb{N}$ is the model dimension, and $Q, K : \mathbb{D}_p{}^d \to \mathbb{D}_p{}^d$ are the query and key affine transformations.

This assumption depends only on the parameters of the query and key affine transformations. It generally holds for most trained Transformer models.

**Theorem 5.2** (Finiteness and Co-finiteness of Languages Recognized by Transformer Decoder)**.** *Assume that Assumption 5.1 holds. Under this assumption, the languages recognized by any Transformer decoders is exactly finite or co-finite languages. Specifically, the following two statements hold:*

1. **(upper bound)** *For any $p \in \mathbb{N}, t(n) \in \Omega(1), \theta \in$ Params, $F \subseteq \mathbb{V}$, there exists a finite or co-finite language $L_f$ such that $L(\mathrm{TDec}, F) = L_f$.*

2. **(lower bound)** *For any finite or co-finite language $L'_f$, there exist parameters $p' \in \mathbb{N}, t'(n) \in \Omega(1), \theta' \in$ Params, $F' \subseteq \mathbb{V}$ such that $L'_f = L(\mathrm{TDec}, F')$.*

Theorem 5.2 represents a key result of this study. It states that, under the infinity-freeness (Assumption 5.1) and with fixed precision $p$, the class of languages recognized by Transformer decoders aligns exactly with the class of finite and co-finite languages, regardless of the specific parameters, the number of decoding steps, or the set of accept states. Or vice versa, that means when the input length exceeds a certain number, transformer model cannot distinguish the inputs.

The two claims of Theorem 5.2 are proved in § 5.1 and § 5.2, respectively.

### 5.1. Proof of the Upper Bound under Assumption 5.1

**Lemma 5.3.** *Suppose Assumption 5.1 holds. Then there exists an integer $L \in \mathbb{N}$ with the following property:*

*For any two inputs $w, w' \in \Sigma^*$ with $|w|, |w'| \geq L$, the Transformer decoder $\mathrm{TDec}_p{}^t(w \cdot \langle \mathrm{sep} \rangle; \theta)$ produces the same output tokens as it does for $\mathrm{TDec}_p{}^t(w' \cdot \langle \mathrm{sep} \rangle; \theta)$, provided $w$ and $w'$ share the same final character.*

*Proof.* Let us denote the final token of the input as $v \in \Sigma \cup \{\langle \mathrm{sep} \rangle\}$. By Assumption 5.1, we know that for any vectors $y, y' \in (\mathbb{D}_p)^d$, the dot-product $Q(y) K(y')^\top \neq -\inf$. In particular, we can choose constants $\alpha, \beta$ in $(\mathbb{D}_p)^d$ (related to the embedding of $v$) such that the repeated sum

of $\exp(Q(\alpha) K(\beta)^\top)$ over enough positions saturates the $p$-precision range to $+\inf$. Hence we define $L$ to be the minimum length at which this "$+\inf$ sum" occurs in the causal masking scenario.

Let $w$ be any string with $|w| \geq L$. When the decoder at time-step $r$ attends over all previously seen tokens ($\langle \mathrm{sep} \rangle$ appended at the end), the *softmax denominator* in $\mathrm{Attn}(q_v, K_w, V_w)$ accumulates

$$\sum_{j=1}^{|w|+1} \exp\big(q_v K_j^\top\big)$$

and by the definition of $L$, this sum diverges to $+\inf$ in $p$-precision. Consequently, the fraction $\exp\big(q_v K_{|w|}^\top\big) / (+\inf)$ is effectively $0$ in $p$-precision, making the final token's contribution vanish. Repeating this for each layer (and for each of the $t(|w|)$ auto-regressive decoding steps) shows that any distinct differences in $w$ vs. $w'$ (*provided* their last character is the same) are overshadowed as $|w| \to \infty$.

Thus if $w, w' \in \Sigma^*$ both have length $\geq L$ and share the same last symbol $v$, the decoder outputs $\mathrm{TDec}^t(w \cdot \langle \mathrm{sep} \rangle; \theta)$ and $\mathrm{TDec}^t(w' \cdot \langle \mathrm{sep} \rangle; \theta)$ coincide. In other words, once the input length is beyond $L$, the model cannot further distinguish among long strings ending in the same symbol. $\square$

**Why Lemma 5.3 implies finite/co-finite recognition.** By Lemma 5.3, all strings of length $\geq L$ that share a final character are mapped to the *same* sequence of output tokens under the $t(|w|)$-step decoding. Hence, if for some long string $w$ the decoder *accepts* (i.e. produces a token in $F \subseteq \mathbb{V}$), then *all sufficiently long strings* with the same last letter are also accepted. Thus we obtain either:

- A *co-finite* pattern: the model rejects only finitely many strings (those of length $< L$, plus possibly a few last-letter classes among the long strings), so $L(\mathrm{TDec}_p^t(\cdot; \theta), F)$ is co-finite.

- A *finite* pattern: the model accepts only finitely many cases (if it rejects all length $\geq L$ strings except perhaps a handful).

In both cases, the recognized language is either finite or co-finite.

**Remark on $\langle \mathrm{sep} \rangle$.** Including a special terminal token $\langle \mathrm{sep} \rangle$ in the input helps ensure that the "last symbol" alignment is explicit. Without it, one might rely on actual last letters in $\Sigma$, and the argument becomes a suffix-based distinction rather than a crisp boundary. Our Definition 4.3 ensures that $w \langle \mathrm{sep} \rangle$ standardizes the final token (*or* the last letter in $w$ if no $\langle \mathrm{sep} \rangle$ is appended), leading to a simpler classification at large lengths.

## 5.2. Proof of the Lower Bound under Assumption 5.1

In this subsection, we show that a Transformer decoder can recognize *any* finite or co-finite language $L_f \subseteq \Sigma^*$ in *constant* (one or two) decoding steps. Formally, we will construct a $p$-precision Transformer decoder that outputs a special "accept" token (e.g. $\langle eos \rangle$) if and only if the input string belongs to $L_f$.

**Lemma 5.4.** *Let $L_{fin} \subseteq \Sigma^*$ be any finite language. Then there exist: a precision parameter $p \in \mathbb{N}$, a parameter set $\theta \in \mathrm{Params}$, and a set of accept state tokens $F \subseteq \mathbb{V}$, such that the decoder recognizes $L_{fin}$ in exactly* one *decoding step. That is, $L(TDec, F) = L_{fin}$.*

*Proof.* We design a two-layer Transformer decoder that first (*i*) accumulates sufficient information (e.g. a partial sum or isomorphic encoding of the entire input string), and then (*ii*) employs a feed-forward network (FFN) to map that information to a binary output: namely "$w \in L$" or "$w \notin L$". When $w \in L_{fin}$, the decoder emits a special token ($\langle eos \rangle$) on the single decoding step; otherwise it does not.

**Embedding layer.** Suppose the input tokens are $w_1, \ldots, w_n \in \Sigma$. Let $p$ be large enough to accommodate all numerics (we will specify $p$ in a moment). For each token $w_i$, define its embedding vector as

$$\mathbf{x}_i := [0, \mathrm{emb}(w_i)] \in \mathbb{D}_p{}^d$$
$$\text{where } \mathrm{emb}(w_i) \in \mathbb{D}_p{}^{d-1}. \tag{3}$$

The extra leading coordinate ($0$) will be used to store positions or partial sums in subsequent layers.

**First attention layer.** We apply a *uniform attention* to gather position-related or partial-sum information. For instance, let the query, key, and value transformations be:

$$Q(\mathbf{x}) = \mathbf{1}, \quad K(\mathbf{x}) = \mathbf{1},$$
$$V(\mathbf{x}) = [1, 0, \ldots, 0] \in \mathbb{D}_p{}^d \tag{4}$$

for all input vectors $\mathbf{x}$. Then, under causal masking (each $\mathbf{x}_i$ only attends to $\mathbf{x}_{1:i}$), the attention output for $\mathbf{x}_i$ is:

$$\mathrm{Attn}\big(Q(\mathbf{x}_i), K(\mathbf{x}_{1:i}), V(\mathbf{x}_{1:i})\big) = \left[\tfrac{1}{i}, 0, \ldots, 0\right]. \tag{5}$$

Thus, after adding the residual connection, the layer output becomes:

$$\mathbf{a}_i^1 = \left[\tfrac{1}{i}, 0, \ldots, 0\right] + \mathbf{x}_i = \left[\tfrac{1}{i}, \mathrm{emb}(w_i)\right]. \tag{6}$$

**First feed-forward network.** We design an FFN so that

$$\mathbf{z}_i^1 = \mathrm{FFN}\big(\mathbf{a}_i^1\big) = \left[0, \tfrac{\mathrm{emb}(w_i)}{i}\right] \tag{7}$$

This step ensures each position's embedding is scaled by $1/i$ and placed in the tail part of the vector.

**Second attention layer.** We next use the $n$-th token $\mathbf{x}_n$ (or similarly the "final step") to attend over all $\mathbf{z}_1^1, \ldots, \mathbf{z}_n^1$. Let $Q(\mathbf{x}) = \mathbf{1}$, $K(\mathbf{x}) = \mathbf{1}$, $V(\mathbf{x}) = \mathbf{x}$. Hence,

$$\mathbf{a}_n^2 = \mathrm{Attn}\big(Q(\mathbf{z}_n^1), K(\mathbf{z}_{1:n}^1), V(\mathbf{z}_{1:n}^1)\big)$$
$$= \tfrac{1}{n}\left[0, \sum_{k=1}^{n} \tfrac{\mathrm{emb}(w_k)}{k}\right] \tag{8}$$

Since we choose $p$ large enough, $\frac{1}{n} \neq_p 0$ in the $p$-precision sense.

**Remark.** The partial sum $\sum_{k=1}^{n} \frac{\mathrm{emb}(w_k)}{k}$ can be seen as carrying isomorphic information about $(w_1, \ldots, w_n)$, assuming a suitable injection or universal approximation property (we treat details abstractly here).

**Second feed-forward network.** Finally, we use a universal-approximation argument: there is an MLP or FFN that can decode $\mathbf{a}_n^2 \sim w_{1:n}$ and output 1 iff $w \in L_{fin}$:

$$\mathrm{FFN}^2\big(\mathbf{a}_n^2\big) = \begin{cases} 1 & (\text{if } w \in L_{fin}), \\ 0 & (\text{otherwise}). \end{cases} \tag{9}$$

We then interpret output "1" as a special accept token (e.g. $\langle eos \rangle$) in the output layer. Hence, the entire decoder recognizes exactly $L_{fin}$ $\qquad \square$

**Extension to co-finite languages.** For a co-finite language $L_{cofin}$, we simply invert the behavior: almost all strings map to "1" (accept), while the finite exceptional set $\Sigma^* \setminus L_{cofin}$ maps to "0." A parallel argument with slight modifications (where the second FFN outputs 1 for nearly all inputs, except a finite listed set of strings) completes the proof.

**Conclusion.** By combining these constructions, we see that any finite or co-finite language $L_f \subseteq \Sigma^*$ can indeed be recognized by a $p$-precision, two-layer Transformer decoder *in one or two decoding steps*. Thus, for such $L_f$, we have $L_f = L(TDec)$ for some parameter choice and constant decode budget.

In summary, *Assumption 5.1* plus the no-positional-encoding policy forces the Transformer decoder to unify all sufficiently long strings with identical trailing tokens. Hence the language recognized cannot exceed finite or co-finite sets, completing the proof of the upper bound.

## 6. Main Result 2: The language recognized by fixed-precision decoder

### 6.1. Lower Bound for asm. 5.1 and APE

We now prove that a Transformer decoder *with Assumption 5.1 and some APE* can recognize any *cyclic language*.

**Theorem 6.1.** *For any $m$-cyclic language $L_c$, there exist some Transformer with asm. 5.1 and Abusolute Positional Encoding,* TDec$'$ *such that $L_c = L(\text{TDec}', \text{F}_c)$ for some set of special tokens $F_c \subseteq \mathbb{V}$*

*Proof Sketch.* **APE to distinguish positions mod $m$** For given $m$-cyclic language, prepare suitable APE such that have periodicity (e.g., sinusoidal embeddings (Vaswani et al., 2017; Chiang et al., 2023)), and the Transformer can effectively identify each position's residue class modulo $m$ Hence, for index $i$ and $j$, if $i \equiv j \pmod{m}$, their positional encodings can be made same so that the network recognizes the same residue class.

**Attention mechanism** In other words, for the head corresponding to residue $r$, only tokens $x_i$ with $i \equiv r \pmod{m}$ receive a high attention score. Under the "inf-free" condition, no key–query product becomes $-\inf$, so we can rely on softmax-based attention to highlight precisely those tokens that belong to the right residue class.

**FFN to implement the $m$-cyclic condition** Since an $m$-cyclic language determines acceptance based on how symbols appear in these residue classes, the final feed-forward network can be crafted to check the patterns aggregated from each class. Concretely, if $L_c$ says "Positions $\equiv r \pmod{m}$ must contain letter $a$" or "must exclude letter $b$," then after the multi-head attention, the hidden representation has sufficient information to confirm or deny these constraints.

Putting it all together, the Transformer obtains position-residue awareness from APE, employs attention to gather all tokens of each residue class, and checks with a FFN whether the cyclic criteria are satisfied. Thus, for any $m$-cyclic language $L_c$, we construct a suitable Transformer decoder (satisfying inf-free and using APE) so that $L_c$ is recognized exactly by that model, completing the proof. $\square$

### 6.2. Lower Bound for NoPE: Letter-set Languages

We now prove that a Transformer decoder *without any positional encoding* (NoPE) *and Assumption 5.1* can recognize any *letter-set language*. Formally, a *letter-set language* $L_S \subseteq \Sigma^*$ is one where acceptance only depends on which *letters* (symbols) appear in $w$ (not on their order or count).

**Theorem 6.2.** *For any letter-set language $L_S$, there exist some Transformer decoders with No Positional Encoding and without Assumption 5.1,* TDec$''$ *such that $L_s = L(\text{TDec}'', \text{F}_s)$ for some set of special tokens $F_s \subseteq \mathbb{V}$*

*Proof Sketch.* A letter-set language is determined solely by the set of unique letters in the input string. In a Transformer decoder without positional encoding, identical letters are mapped to identical embedding vectors, irrespective of their positions in the input sequence. Consequently, the model cannot distinguish whether $a$ appears as the first or fifth letter, but it can identify whether $a$ is present in the input at all. By processing embeddings, the model can determine the existence of each letter without tracking its count or position.

Using attention and feed-forward layers, the model can consolidate these embeddings to produce a "presence flag" for each letter. The flag is set based on whether the embedding is zero or non-zero. Thus, we employ the non-finite floating-point value $\inf$ in the denominator during the transition computation to make the flags zero aligning with the discussion in Lamma 5.3. This is why the Assumption 5.1 is removed. For example, if $a$ appears anywhere in the sequence, a specific hidden vector state can be activated to indicate its presence.

Since letter-set languages are defined by finite logical combinations of conditions on letter presence, the final feed-forward and output layers can evaluate these conditions. For example, the model can output an accept token if the presence flags match the required subset $S$, or reject otherwise. This process effectively ignores order and frequency, focusing solely on whether each required letter is present at least once.

The lack of positional encoding aligns naturally with the requirements of letter-set languages. A NoPE Transformer focuses on whether a given letter appears, without being influenced by order or frequency. Even if a letter $a$ appears multiple times, the model only needs a single bit of information ("$a$ exists") to make its decision. By aggregating these presence flags, the Transformer can determine whether the input satisfies the rules of the letter-set language.

Thus, a NoPE Transformer can recognize any letter-set language, using its ability to abstract away positional information and focus on the presence of letters. $\square$

## 7. Discussions

### 7.1. What is the Key Module in Transformers?

Although numerous studies have advanced our understanding of Transformers, a fundamental question remains: *"Which architectural component primarily contributes to their expressivity?"* Despite extensive research on elements like attention mechanisms, layer normalization, and embedding schemes, there is no universal consensus on *what exactly* determines a language model's ability to capture complex linguistic phenomena.

Bhattamishra et al. (2020b) focused on the Turing-completeness and the necessity of various architectural components and highlighted the crucial role of residual connections in maintaining expressivity. They also demonstrated that Transformers without explicit positional encoding but with positional masking remain Turing-complete. Similarly, Chiang et al. (2023) highlighted the importance of numerical precision (fixed vs arbitrary) and showed that the expressivity of such an encoder Transformer can be tightly upper-bounded by the language class FOC[+; MOD], a first-order logic with counting quantifiers, addition, and modular arithmetic.

A crucial difference emerges when comparing their results to ours: they included *positional encoding* (specifically a sinusoidal scheme), which allowed the model to handle periodic information effectively. In this study, We adopted a constant precision scheme similar to Chiang et al. (2023). Moreover, we introduced a reasonable practical Assumption 5.1. Building on these settings,we identified a Transformer setup capable of recognizing the minimal language, namely finite or co-finite languages, without any positional encodings (§ 5). This setup closely resembles real-world Transformers, leading us to hypothesize that practical Transformers may inherently be restricted to recognizing finite languages, functioning as highly efficient lookup tables. While prior studies (Bhattamishra et al., 2020b; Kazemnejad et al., 2024)) demonstrated the practical effectiveness of NoPE, our theoretical analysis suggests that NoPE has inherent limitations in enhancing expressivity.

Next, we examined how adding absolute positional encoding (APE) and removing the assumption affected the tendency to restrict recognition to finite languages (§ 6). However, even with these additions, expressivity increased only slightly, as fixed-precision still constrains expressivity to near-finiteness. Our findings show that restricting precision from logarithmic (Merrill & Sabharwal, 2024b) to constant results in a significant loss of expressiveness. Furthermore, this loss increases as the number of decoding iterations grows, noting that expressivity reaches P when $t \in \mathcal{O}(n^c)$.

### 7.2. Langue Modeling

Throughout this work, we frame the Transformer as a *language recognizer*, addressing the membership problem in a more formal sense rather than as a *language generator*.

In practice, particularly in language modeling, a decoder-based Transformer typically produces tokens probabilistically, generating text rather than deciding membership in a formal language. In fact, research on the expressivity of language modeling exists (Svete & Cotterell, 2024; Nowak et al., 2024). While our "recognizer" viewpoint diverges somewhat from typical usage, bridging these two outlooks more rigorously remains a key objective for future research.

### 7.3. Potential Extensions

We acknowledge that our current setup is simplified, focusing on a limited subset of Transformer components: attention masking, the absence of layer normalization, and no extensive multi-head or multi-layer structure. In real-world architectures, additional architectural features could significantly impact expressivity.

Furthermore, we have identified gaps (Table 1). A natural extension involves clarifying how these additional mechanisms, such as relative positional encoding or the softmax-to-hardmax transition, might shift the upper and lower bounds on expressivity. We believe our fundamental approach can be adapted to investigate such enhancements, while leaving precise formalization and empirical validation for future work.

## 8. Conclusion

In this work, we examined the expressivity of fixed-precision Transformers to investigate their practical implications. To achieve this, we introduced three constraints: fixed-precision floating-point arithmetic, a reasonable assumption 5.1 regarding query-key parameters, and the presence or absence of positional encoding.

In § 5, we demonstrated that Transformers operating under the constraints (Fixed-precision + Assumption 5.1 + NoPE) can recognize only finite or co-finite languages. In § 6, we further proved the role of Assumption 5.1 and also positional encoding, as relaxing either of these constraints slightly enhances expressivity.

These findings suggest that these constraints impose fundamental limitations on Transformer expressivity. Future research could extend this analysis to language modeling or investigate how alternative modules and hardmax replacements influence expressivity.

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
