# OpenReview forum: "The Expressivity of Fixed-Precision Transformers without Positional Encoding"
_ICML.cc/2025/Conference — Submitted to ICML 2025_

### Official Review · Reviewer_VayC · 2025-03-04

**Overall Recommendation:** 1

**Summary:**

The paper explores the expressive capabilities of transformer decoders constrained by a fixed-precision setting, such as a specific floating-point arithmetic, and with limited positional embedding, utilising formal language theory.

Specifically, the paper posits that if a particular assumption concerning the query and key matrices of the examined transformers is satisfied (Assumption 5.1), then transformers devoid of positional embedding recognise only finite or cofinite languages (Section 5), whereas transformers with an absolute positional embedding can recognise all cyclic languages. It is further contended that transformers not conforming to Assumption 5.1 recognise letter set languages.

## update after rebuttal
No changes, I stand with my initial review.

**Claims And Evidence:**

While I do not doubt the general correctness of the results, I find the proofs in this paper unconvincing for a few reasons:

- The precise definition of the transformer decoders considered in this paper is somewhat vague. Although the details are just enough to understand the upper bounds, the lower bounds require more clarification. Specific aspects, such as the type of attention mechanism employed, are not explicitly defined. I felt like these details are spread throughout the paper, but the absence of a succinct definition that encapsulates these elements, along with theorems that clearly state the established results, makes it difficult for me to be certain about the contribution of this paper.

- There are several details that leave me puzzled. For example, in Definition 4.2, what does \sigma represent? I assume it stands for the input word, previously referred to as w. Similarly, in Definition 4.3, what does (sep) stand for? Could it be (EOS)? While these are minor issues, similar ambiguities scattered throughout the main paper impede a comprehensive understanding.

- It strikes me as odd that there are no extensive formal proofs provided. Often proofs seem more like sketches (for example Theorem 6.1) or are distributed across an entire section, which does not aid comprehension. Additionally, there is no technical appendix to support these more informal arguments presented in the paper.

- I'm finding it difficult to grasp the importance of Assumption 5.1. The paper mentions that this assumption "generally holds for most trained transformers," yet I remain sceptical. For instance, in the context of using a transformer in a low-bit setting, it's conceivable that an overflow situation might occur, resulting in reaching either -infty or infty.

**Essential References Not Discussed:**

None that I am aware of.

**Experimental Designs Or Analyses:**

Not applicable.

**Methods And Evaluation Criteria:**

See above regarding my comment on the form of proofs provided.

**Other Comments Or Suggestions:**

None besides those stated above or posed as question below.

**Other Strengths And Weaknesses:**

strengths:
- The paper offers intuitive explanations and informal descriptions wherever feasible.
- Overall, despite crucial technical details, the core contribution of the paper is relatively clear.


weaknesses:
- While it's novel, I find the scenario where a transformer, devoid of positional embedding and constrained by fixed-precision arithmetic, leads to finite or cofinite languages, somewhat predictable. I am not aware of any existing proofs on this exact setting, thus its new, but I am not entirely convinced of its significance.

**Questions For Authors:**

I recognise that my critique regarding the lack of details might be somewhat severe. Could you clarify the exact definitions of:
  - Absolute positional embeddings?
  - What is the precise class of transformers that contribute to the lower bounds you present?

Aside from this:
- Why have you opted to provide only a proof sketch for Theorem 6.1? I am struggling to see how the technical details merit this approach.

**Relation To Broader Scientific Literature:**

This paper is well-placed in an active line of research concerned with understanding the expressive
power of transformers using formal language theory.

**Theoretical Claims:**

I endeavoured to understand the results but found it challenging due to a lack of clarity regarding the specific setting considered in these outcomes.

---

> ### Author Rebuttal · Authors · 2025-04-01
>
> We sincerely thank the reviewer for their thoughtful and constructive feedback. We greatly appreciate your careful reading and insightful suggestions, which have helped us clarify key aspects of the paper. Below, we respond point by point to the concerns raised.
>
> # Weaknesses
> ## 1. Inprecision of definition of Transformer, unclear symbol in definitions
> - We acknowledge that our definitions of Transformer components, as well as our explanations for some symbols (e.g., $\sigma$, <sep>), were insufficient. We appreciate the reviewer pointing this out, and we will revise all relevant parts of the manuscript to provide more precise and consistent definitions throughout.
>
> ## 2. The validity and importance of Assumption 5.1
> - validity
> 	- We agree that Assumption 5.1 may not always hold—particularly in low-bit settings. However, our intention was to model more typical settings found in practical scale models such as BERT or LLaMA, where floating-point precision (e.g., FP16 or FP32) is generally sufficient.
> 	- In cases where precision is severely limited, we expect this assumption to break down, which is precisely why we chose to state it explicitly as an assumption.
> - importance
> 	- The reviewer notes that a Transformer without positional encodings and with fixed-precision arithmetic yielding only finite or cofinite languages may appear somewhat expected. However, this outcome does not follow from those conditions alone. As shown in Table 1, even under fixed precision and NoPE, the expressivity exceeds FinCofin unless Assumption 5.1 is also imposed. With this assumption, the expressivity becomes exactly FinCofin, both as an upper and lower bound.
> 	- In this sense, Assumption 5.1 plays a critical role in identifying a tight boundary on expressivity. We view our contribution as providing a theoretical grounding for how fixed-precision constraints shape the model’s representational limits and establishing FinCofin as the precise limit under minimal yet meaningful assumptions.
>
> # Question for Authors
> ## 1. (Clarification) Absolute positional embeddings
> - In our setting, we treat APE as a function of the form $\mathrm{APE}: \mathbb{N} \to D_p^d$ which maps each position to a $d$-dimensional vector in fixed-precision float. This encoding is added element-wise to the token embedding before being passed into the Transformer block.
> 	- Note that under this definition, learnable positional embedding, such as those used in GPT-2—are considered a subclass of APE. We will revise the manuscript to clearly include this definition.
>
> ## 2. (Clarification) Transformers which contribute to the lower bound
> - We describe the architecture used to establish the lower bound in Section 4.4. This includes the model’s structure and computational setting. (please ignore "and attention head" in line 189)
> - If there are specific aspects that were unclear or insufficiently detailed, we would greatly appreciate your guidance on which parts should be clarified. We are happy to revise the manuscript accordingly.
>
> ## 3. Theorem 6.1 will be exclued
> - We sincerely thank the reviewer for raising this important issue. We agree that the current proof sketch for Theorem 6.1 lacks sufficient formal detail, and that the argument, as presented across Section 6.1, does not aid comprehension.
> 	- Due to time constraints during the submission process, we included only a high-level idea without a fully formalized proof.
> 	- We found that the lower bound for APE is, in fact, no stronger than that for NoPE under our current assumptions. While a tighter lower bound may exist, it likely requires a different construction and falls outside the scope of this work.
> - Accordingly, we plan to exclude Section 6.1 entirely from the final version. We will instead mention this direction briefly as a possible avenue for future research. We believe this change will improve the overall clarity and integrity of the paper.
>
> # Summary
> - Once again, we are grateful for your detailed review and recognition of the core contributions of our work. We believe the revisions we plan to make based on the comments. Particularly regarding the clarity of definitions, the role of key assumptions, and the exclusion of Section 6.1.
> - Please feel free to let us know if there are further points that would benefit from clarification.

---

### Official Review · Reviewer_PFKJ · 2025-03-13

**Overall Recommendation:** 3

**Summary:**

The authors study the expressivity of transformers while considering the practical constraints of real-world usage, such as fixed-point precision. The authors show that without positional encoding, transformers can only represent finite and co-finite languages, which are subclasses of regular language. Adding positional encoding improves the expressivity but does not alleviate the main limitations due to fixed-point precision. The authors conclude with a detailed discussion, highlighting the gap between theoretical models and models used in practice. Their analysis leads to the conclusion that transformers, because of fixed-point precision and other practical limitations, are akin to very efficient look-up tables that can express finite and co-finite languages.

**Claims And Evidence:**

The theoretical claims are supported by clear and detailed proofs.

**Essential References Not Discussed:**

To the best of my knowledge, there were no essential references not discussed in the current submission.

**Experimental Designs Or Analyses:**

There were no experiments conducted.

**Methods And Evaluation Criteria:**

The authors provide theoretical results to better understand the expressivity of transformers. The model is simplified to remain in a controlled setting, and the method seems to make sense for the problem at hand.

**Other Comments Or Suggestions:**

I list below some potential typos:
- Table 1: "Uppoer" --> "Upper"
- Table 1 caption: "positinoal" "positional"
- l. 119, second column: "theoretic" --> "theoretical"
- l.165: Given that GPTs use causal attention, should it be "Since decoder-only Transformer" instead of "Since encoder-only Transformer"?
- l.167: "absolute" --> "absolute"
- l.165: "generation," --> "generation."
- l.400: ", In this study, We adopted" --> "In this study, we adopted"
- l. 426, first column: "Languge Modeling" --> "Language Modeling"

**Other Strengths And Weaknesses:**

**Strengths**

- The paper is well-written, and the problem tackled seems of great interest
- The findings are contextualized and explained to grasp the main intuitions
- I appreciate that the authors consider a setting as realistic as possible
- I appreciate the discussion and "possible extensions" parts that list the limitations of the current submission and potential directions for future work. This helps understand the contributions of the submission without over-stating them

**Weakness**

I list below what I think are weaknesses, but I would be happy to be corrected if I misunderstood some important aspects of the authors' contributions.

- Some technical parts on the formal language part are not well-detailed, notably the FOC[+; MOD] ones, which makes it hard to grasp all the impact of the submissions' contributions for non-experts
- Most transformers used in practice, notably in language tasks, use learnable positional encoding or other more involved frameworks (e.g., ROPE [1]) that consist of causal attention with decoder-only blocks. In the current simplified setting of the submission, it is not clear, at least to me, how the insights would translate to those more common models.
- It is not clear, at least to me, if the submission conclusion is informative of what we observe in practice, that is, that transformers show surprising generalization and emerging capabilities. I am doubtful that efficient look-up tables would match such capabilities, but I would be happy to hear what the authors think of that.

Overall, I find the paper well-written, and the contributions seem valuable, although the setting considered is simplified. This is the reason for my score, but I am ready to modify it depending on the discussion with the authors.

*References*

[1] Su et al. RoFormer: Enhanced Transformer with Rotary Position Embedding, arXiv 2023

**Questions For Authors:**

1) In practice, most transformers make use of positional encoding, either deterministic or learnable or more complex ones (ROPE). How would the insights translate to such a model given that they can mitigate the permutation-onvariance of attention modules?

2) Same question for transformers used in practice, notably in language tasks, that use causal attention in decoder-only blocks.

**Relation To Broader Scientific Literature:**

I believe that the related work and prior works are well introduced and compared. The submission's contributions are interesting and are part of a growing interest in the literature to better understand the expressivity of transformers. The novelty seems to be in the analysis from a formal language perspective with realistic assumptions on the models, notably the fixed-point precision one which is indeed a real limitation in practice.

**Theoretical Claims:**

The theoretical findings are supported by detailed and clear proofs.

---

> ### Author Rebuttal · Authors · 2025-04-01
>
> Thank you very much for your detailed and thoughtful review.
> We are encouraged by the recognition of key strength and we appreciate the thoughtful suggestions for improvement.
> We greatly appreciate your recognition of the strengths of our work, as well as your thoughtful suggestions for improvement.
>
> # Weaknesses
> ## 1. On the lack of explanation for FOC[+; MOD] and related language classes:
> -  We agree with the reviewer that Table 1 refers to FOC[+; MOD], but the manuscript does not include even a brief explanation. This is an important oversight, especially for non-experts. We will revise the paper to include concise definitions and necessary backgrounds so that the impact of our results can be better understood. Thank you for pointing this out.
>
> ## 2. On the applicability of our results to more realistic Transformer architectures (e.g., with learnable PE or RoPE):
> - We address this concern in more detail under the “Questions for Authors” section below.
>
> ## 3. On the practical relevance of our conclusions, especially with respect to generalization and emergent capabilities:
> - As also noted in our response to Reviewer Je32, we acknowledge that the term “lookup table” was overstated. While valid in a theoretical sense under fixed precision, it does not fully capture a model’s generalization capacity. We view expressivity and generalization as distinct: even under strict constraints, a model can still generalize within its representable class.
> - Although our results do not directly explain emergent behaviors in large-scale Transformers, they provide a foundation for understanding expressivity under resource constraints—an important step toward formalizing learnability. We hope our simplified framework helps clarify how architectural choices affect expressivity and contributes to bridging theory and practice.
>
> # Other Comments Or Suggestions
> - We also thank the reviewer for pointing out minor typos and expressions a lot that could be improved.
> - > l.165: Given that GPTs use causal attention, should it be "Since decoder-only Transformer" instead of "Since encoder-only Transformer"?
> 	- As noted in our response to Reviewer Je32, our intention was indeed to contrast the decoder-only setup with encoder-based models, but the sentence as written was confusing. We will revise the text to clarify our intended meaning.
>
> # Questions For Authors
> ## 1. The effects of positional encodings for the permutation-invariance of attention modules
> > In practice, most transformers make use of positional encoding, either deterministic or learnable or more complex ones (ROPE). How would the insights translate to such a model given that they can mitigate the permutation-invariance of attention modules?
> - We appreciate this important question regarding the extension to more realistic positional encoding schemes. In fact, learnable positional encodings were implicitly treated in our framework as a form of absolute positional encoding (APE). In the manuscript, we focused on periodic APE as a representative idealized case.
> - As for RoPE, it introduces a structurally distinct behavior compared to NoPE and APE. Although a detailed analysis of RoPE is outside the scope of this work and would require separate treatment, we are also interested in this direction and plan to investigate RoPE-based architectures in future work.
>
>  ## 2. Decoder-Only Transformers and Expressivity
>  > Same question for transformers used in practice, notably in language tasks, that use causal attention in decoder-only blocks.
>   - Regarding decoder-only architectures, our setting assumes $\Omega(1)$ decoding steps. Therefore, our results apply to any number of decoding steps that scale with the input length.
> 	- Importantly, our results show that in fixed-precision settings, the number of decoding steps does not affect the model’s expressivity.
> 	- In this context, causal attention enables NoPE Transformers to infer the relative positions of input tokens, partially compensating for the lack of explicit positional encodings.
> - That said, we recognize that generalizing our results to broader architectural variations remains a challenging and important direction. In future work, we hope to extend the framework to cover other types of positional encodings and as well as additional architectural features such as sparse attention or grouped-query attention.
>
> # Summary
> - We again thank the reviewer for their thoughtful feedback—including the helpful identification of typos, and kind recognition of our work’s strengths. Your comments have helped us improve both the clarity and positioning of our contributions.
>
> - While our setting is simplified, we believe it offers a useful foundation for understanding expressivity under resource constraints, and we will revise the manuscript to better explain technical details and clarify connections to practical models.

---

> > ### Comment · Reviewer_PFKJ · 2025-04-06
> >
> > I thank the authors for their detailed answers and for their efforts to address my concerns. I will consider that along with the other reviews for my final recommendation.

---

### Official Review · Reviewer_zhee · 2025-03-14

**Overall Recommendation:** 2

**Summary:**

This paper investigates the expressivity of the transformer architecture when it is constrained to operate at fixed numerical precision and not involve any infinite parameters -- a setup seemingly matching real-world setups closely. Perhaps surprisingly, the results indicate that the architecture can only recognize finite and co-finite formal languages, suggesting that the architecture is ultimately limited to finite memorization.

## update after rebuttal

I stand by my original assessment that the relevance to real-world models remains unclear.

**Claims And Evidence:**

I believe that the mathematical theorems in the paper are correct.

However, I am unsure about the implications for machine learning that the paper claims. The abstract suggests that practical transformers “effectively function[] as little more than efficient lookup tables”. This conclusion seems entirely at odds with a variety of work finding transformers to empirically show nontrivial generalization, e.g. length generalizing in algorithmic tasks [1,2,3] or generalizing to unseen in-context learning tasks [4]. While real-world transformers undeniably operate in fixed precision, it remains unclear how useful this asymptotic is for understanding empirical behavior. Note that given plausible precision and size of real-world transformers, the sizes of the finite languages that they can potentially model will still be extremely large, large enough to lead to nontrivial algorithmic generalization at finite lengths.
Experiments backing up the relevance of the conclusions to actually implemented machine learning at reasonable input lengths would be a way of helping make the claim that transformers, due to the theorems proven here, are indeed "little more than efficient lookup tables".


[1] Zhou et al, What Algorithms can Transformers Learn? A Study in Length Generalization, ICLR 2024
[2] Kazemnejad et al, 2024 (cited in the paper)
[3] Huang et al, A Formal Framework for Understanding Length Generalization in Transformers, ICLR 2025
[4] Garg et al, What Can Transformers Learn In-Context? A Case Study of Simple Function Classes, http://arxiv.org/abs/2208.01066

**Essential References Not Discussed:**

I believe all key references are discussed.

**Experimental Designs Or Analyses:**

N/A (no empirical experiments).

**Methods And Evaluation Criteria:**

N/A (no empirical evaluation).

**Other Comments Or Suggestions:**

Clarification: line 142 (left) “such as“: are any other special values than +inf and -inf included?

Line 140 (right column): “there is no intersection to alphabet in this study” – I don’t understand what this means, can the authors clarify?

Line 231: “holds for most trained transformer models” – why “most” and not “all”? In real-world transformers, aren’t all weights finite?

**Other Strengths And Weaknesses:**

In my view, the primary weakness is that, as described under "Claims and Evidence", it is unclear whether the strong limitations (effectively precluding generalization to unbounded-length inputs) are relevant at all to real-world training of transformers. After all, one could just as well conclude that standard computers (which, grounded in the physical world, have finite RAM) can only compute finite-state languages, whereas there is a general consensus that real computers are Turing-complete for most practical purposes. Thus, in the absence of experiments or further non-asymptotic results, the relevance of the results to machine learning remain unclear.

**Questions For Authors:**

How do the results relate to results relating finite-precision transformers to C-RASP [5] and AC0 [6], which are far larger classes than FinCofin? Do the results from the present paper subsume those, or are there differences in the details of the formalization?

[5] Yang and Chiang, Counting Like Transformers: Compiling Temporal Counting Logic Into Softmax Transformers, COLM 2024
[6] Feng et al, How Numerical Precision Affects Mathematical Reasoning Capabilities of LLMs, arxiv 2024

**Relation To Broader Scientific Literature:**

The paper links up to a recent line of work on the theoretical expressiveness of the transformer architecture. It is already known that transformers with finite precision face strong limitations (Proposition 1 in Merrill&Sabharwal 2024a, cited in the paper). Whereas that reference took this result as evidence to look for other theoretical models, the current paper strengthens this result in the NoPE setup by showing that in fact the expressivity equals the class FinCofin.

**Theoretical Claims:**

The theorems appear sound to me. I believe the theorems to be correct based both on my reading of the proofs and on my experience in this subfield.

---

> ### Author Rebuttal · Authors · 2025-04-01
>
> We are grateful to the reviewer for the constructive and insightful feedback.
>
> # Weaknesses:
> We thank the reviewer for pointing out the apparent contradiction between our claim ("fixed-precision Transformers behave as efficient lookup tables") and the observed generalization in experiments. We acknowledge that the phrase "lookup table" in the abstract may have been misleading.
> Our intent was to highlight theoretical limitations under fixed precision, not to imply literal memorization. Our analysis suggests that when precision is insufficient, fixed-precision Transformers cannot approximate the infinite-precision behavior seen in prior work.
> While we did not fully address what constitutes "sufficient" precision in the current manuscript, we plan to explore this in future work, as the reviewer helpfully suggested.
>
> Although it may seem trivial, our contribution is to understand how architectural choices within the Transformer family influence the classes of languages that can be represented—even under the same finite-precision assumptions as a boundary analysis that explicitly identifies how changes—such as removing or adding architectural components—impact expressivity.
>
> # Clarifications
> We thank the reviewer for these helpful clarification requests. Below, we address each of the reviewer's questions in turn.
>
> 1. Line 142 - Special values in floating-point numbers:
> 	- Any other special values except $+\infty$ and $-\infty$?
> 		- In our manuscript, we only intended to refer to the explicitly mentioned values: $+\infty$, $-\infty$, and $\mathrm{nan}$.
> 	- Within the scope of our theoretical setting, NaNs arising from division in the softmax are excluded by Assumption 5.1. However, we acknowledge that other operations (e.g., $0 + \infty$, $0 \times \infty$) may also result in NaNs and should be mentioned explicitly. We thank the reviewer for bringing this to our attention.
>
> 2. Line 140 - Clarification of the phrase, "there is no intersection to alphabet in this study":
> 	- What we intended here is the condition $\Sigma \cap \mathbb{V} = \emptyset$, meaning that the set of special tokens does not intersect with the standard alphabet.
> 		- This restriction was made for clarity: special tokens have distinct roles — e.g., <eos> as an acceptance marker, <sep> as a separator—and keeping them disjoint from the alphabet avoids ambiguity in our definitions.
>
> 3. Line 231 - Assumption 5.1 “holds for most trained transformer models” – why “most” and not “all”:
> 	- This statement refers specifically to the inner product $q(x)k(y)^\mathsf{T}$ for all token pairs $x, y$. While $q(x)$ and $k(y)$ are practically finite in real-world models, it is difficult to make a definitive claim about all possible products.
> 		- Categorically asserting this assumption would require making claims about learned parameters, and we cannot theoretically rule out the possibility that training may produce pathological configurations.
> 	- Furthermore, violations of this assumption could lead to degenerate behaviors such as attention masking or unique hard attention. In contrast to soft attention—where weights are smoothly distributed—hard attention focuses all weight on a single token. Given prior literature, we believe it is important to distinguish soft attention from such extreme cases.
>
> # Question For Authors
> > Does your result subsume the results on C-RASP or AC0? Or is the formalism different?
> - We thank the reviewer for this insightful question, which touches on the relationship between our results and prior work on formal characterizations of Transformer expressivity, particularly in the context of C-RASP and AC0.
>
> 	- Regarding C-RASP, we are aware that the class FOC[+; MOD] is strictly weaker than C-RASP, as discussed in [5]. As we mentioned at table 1, FOC[+; MOD] is one of the upper bound of the expressivity of fixed-precision Transformer models with Absolute PE ([7] Theorem 2, [5] Lemma 6.1, Theorem 7.1)
> 		- Therefore, while our formalism differs, we believe our focus on the weakest setting are complementary.
>
>  - As for AC0, we thank the reviewer for introducing [6], which we were not previously aware of.
> 	 - Their framework, CoT[T(n), d(n), s(n)], characterize expressivity based on decoding steps, embedding size, and precision. While they analyze cases like CoT[log n, poly(n), 1] $\subseteq$ AC0 (Theorem 3.1), our setting corresponds to CoT[T(n), 1, 1], which is not covered in [6]. We show that this setting has expressivity exactly FinCofin — likely strictly weaker upper bound than the classes discussed in their work. Importantly, we also provide a lower bound, which they do not.
>
> # Summary
> We thank the reviewer again for their insightful feedback, which has helped improve the clarity and focus of our work. We hope our responses adequately address the concerns, and we will incorporate the necessary revisions in the final version.

---

> > ### Comment · Reviewer_zhee · 2025-04-02
> >
> > I thank the authors for these helpful clarifications and insightful responses re [5,6] (I didn't find where the authors give the reference for "[7]" -- I trust this will be included in the next version of the paper). Overall, making these aspects clearer will improve the paper.
> >
> > Simultaneously, I do stand by my original assessment that
> > > In my view, the primary weakness is that, as described under "Claims and Evidence", it is unclear whether the strong limitations (effectively precluding generalization to unbounded-length inputs) are relevant at all to real-world training of transformers. After all, one could just as well conclude that standard computers (which, grounded in the physical world, have finite RAM) can only compute finite-state languages, whereas there is a general consensus that real computers are Turing-complete for most practical purposes. Thus, in the absence of experiments or further non-asymptotic results, the relevance of the results to machine learning remain unclear.
> >
> > While I consider this a weakness, I do not think it necessarily precludes publication.

---

### Official Review · Reviewer_Je32 · 2025-03-16

**Overall Recommendation:** 2

**Summary:**

This paper demonstrates that fixed-precision Transformer decoders without positional encoding are limited to recognizing only finite or co-finite languages, and modest expressivity gains are made when adding positional encodings or relaxing parameter constraints. By performing these expressivity analyses in less idealized conditions, the authors address the significant gap between theoretical models and practical implementations, and ultimately suggest that real-world Transformers may be limited in their expressivity (acting effectively as lookup tables).

**Claims And Evidence:**

The main claims in the submission are supported by mathematical proofs. However, as the authors note, the relevance of these claims to real-world settings still needs to be investigated further.

**Essential References Not Discussed:**

There are no essential references missing that I'm aware of.

**Experimental Designs Or Analyses:**

The analyses appear well designed.

**Methods And Evaluation Criteria:**

The mathematical approaches make sense for the problem at hand.

**Other Comments Or Suggestions:**

Please see the questions below.

**Other Strengths And Weaknesses:**

Other strengths include a novel approach to analyzing transformer expressivity and a step towards more practically relevant analysis in this domain.

Other weaknesses include the fact that the proofs seem to lack some rigor, and there's no supplementary material to address this.

**Questions For Authors:**

1) Line 165: why mention encoder only transformers here, which are not the object you study (decoder only transformers)?

2) Is sigma the input in definition 4.2? If so, should it have a time superscript?

3) Assumption 5.1 mentions plus or minus infinity cannot be reached, but the proof of Lemma 5.3 mentions that positive infinity is reached, and it uses Assumption 5.1. This seems contradictory, maybe it's a typo in the assumption.

4) In Lemma 5.3, if “the final token’s contribution vanishes”, then why is it important to have the same final character for w and w’?

5) Line 328: I think “lower” not “upper” bound is meant.

**Relation To Broader Scientific Literature:**

Previous research established that ideal Transformers with infinite precision are Turing-complete, while those with logarithmic precision fall within specific circuit complexity classes like TC$^0$. This paper extends these findings by revealing how practical constraints limit expressivity, highlighting that the theoretical power of Transformer models may be unattainable in actual implementations due to fixed-precision limitations.

**Theoretical Claims:**

I did not carefully check the correctness of proofs and theoretical claims, but they seemed reasonable.

---

> ### Author Rebuttal · Authors · 2025-04-01
>
> We greatly appreciate the valuable comments provided by reviewer Je32.
>
> # Weakness: rigorous of proofs
> We agree with the reviewer’s concern regarding the rigor of our proofs.
> In the current manuscript, we prioritized clarity of explanations. However, we also acknowledge that the proofs are not sufficiently rigorous in the current form. Thus, we fully intend to include additional proof details and supplementary lemmas —particularly in Section 6— in the final version. These additions will ensure that the mathematical rigor meets the standards expected by the community.
>
> # Questions For Authors
> ## 1. Line 165 – Mention of Encoder-Only Transformers:
> > Line 165: why mention encoder only transformers here, which are not the object you study (decoder only transformers)?
> - Our intention in mentioning encoder-only Transformers was to highlight that even in a decoder-only setting (particularly when using NoPE), positional information is implicitly captured through causal masking [1]. The contrast with encoder-based models was intended to underscore our motivation for adopting NoPE.
> - We will therefore revise the text to explicitly clarify our rationale. We sincerely thank the reviewer for bringing this to our attention.
>
> ## 2. $\sigma$ in Definition 4.2:
> > Is sigma the input in definition 4.2? If so, should it have a time superscript?
> - We acknowledge that we omitted an explicit statement clarifying that $\sigma \in \Sigma$. We thank the reviewer for pointing out this oversight and will clearly specify it in the revised manuscript.
>
> - Regarding the suggestion to use a time superscript, we apologize but did not fully understand the reviewer's intent. If the reviewer was referring to a notation like $\sigma_{1:t}$, we would like to clarify that our current definition is intentionally formulated so that such notation is unnecessary, as the autoregressive nature of the definition implicitly encodes this temporal dependency. However, if we have misunderstood the reviewer's suggestion, we would greatly appreciate further clarification.
>
> ## 3. Assumption 5.1 vs. Lemma 5.3:
> > Assumption 5.1 mentions plus or minus infinity cannot be reached, but the proof of Lemma 5.3 mentions that positive infinity is reached, and it uses Assumption 5.1. This seems contradictory, maybe it's a typo in the assumption.
> - Assumption 5.1 asserts that $qK \neq \pm \mathrm{inf}$. Thus, the divergence of the sum $\sum \exp(qK)$ to positive infinity does not contradict this assumption.
>
> ## 4. Lemma 5.3 - the final token's contribution:
> > In Lemma 5.3, if “the final token’s contribution vanishes”, then why is it important to have the same final character for w and w’?
> - The phrase “the final token’s contribution vanishes” was indeed a misstatement. Our intended claim was that the contributions of all tokens *except* the final token vanish. We sincerely thank the reviewer for highlighting this error, and we will correct the statement in the revised manuscript.
>
> ## 5. Line 328 - mistake:
> > Line 328: I think “lower” not “upper” bound is meant.
> - We agree with the reviewer’s observation that “lower bound” is the appropriate term. This error will be corrected in the revised version. We appreciate the reviewer pointing this out.
>
> # Summary
> Overall, while the current manuscript prioritizes clarity and simplicity in its presentation, we will carefully incorporate the reviewer’s suggestions, including the mentioned clarifications and additional proof details, to enhance the rigor of our work.
> We sincerely thank the reviewer once again for these insightful and constructive comments, which have significantly helped us improve the quality of the paper.
>
> We hope that our responses sufficiently address the reviewer’s concerns and clarify the contributions of our work. Please let us know if any further details or clarifications would be helpful.

---

> > ### Comment · Reviewer_Je32 · 2025-04-03
> >
> > Thank you for the helpful response! I will consider this further along with the other reviews (and their responses) to settle on a final recommendation.
> >
> > This is minor, but the fact that $\sigma \in \Sigma$ wasn't the confusing part. I was confused by how the transformer input could be represented as $\sigma$ concatenated with just the prior timestep's output. I would think it would have to be concatenated with all prior outputs.

---

> > > ### Author Response · Authors · 2025-04-08
> > >
> > > Thank you. Now I understand what you say. It's true that my definition doesn't allow reference to past outputs. I'll improve it later.

---

### Decision · Program_Chairs · 2025-05-01

**Decision:**

Reject

**Comment:**

This paper studies the expressivity of the transformer architecture when it is restricted to operate at fixed numerical precision and not involve any infinite parameters. It is shown that such a restricted architecture can only recognize finite and co-finite formal languages, and that gains made via positional encodings and relaxations of parameter constraints are relatively small. This is used to argue that real-world transformers are ultimately limited in their expressiveness.

The paper could be an interesting addition to the literature on the expressiveness of transformers. However, the proofs are somewhat sloppy, and the paper doesn't have an appendix -- a problem for a paper whose contributions are primarily foundational. Also, I agree with several of the reviewers that claims about these results' real-world relevance ought to be toned down. Given this, I am recommending rejection. I encourage the authors to develop a more polished version of the manuscript that incorporates the reviewers' feedback and submit it to a different venue.